# Preparation of Metal-Immobilized Methacrylate-Based Monolithic Columns for Flow-Through Cross-Coupling Reactions

**DOI:** 10.3390/molecules26237346

**Published:** 2021-12-03

**Authors:** Akhmad Sabarudin, Shin Shu, Kazuhiro Yamamoto, Tomonari Umemura

**Affiliations:** 1Department of Chemistry, Faculty of Science, Brawijaya University, Jl Veteran, Malang 65145, Indonesia; 2Graduate School of Engineering, Nagoya University, Furo-cho, Chikusa-ku, Nagoya 464-8603, Japan; shu.shin@c.mbox.nagoya-u.ac.jp (S.S.); kazuhiro@mech.nagoya-u.ac.jp (K.Y.); 3School of Life Sciences, Tokyo University of Pharmacy and Life Sciences, 1432-1 Horinouchi, Hachioji 192-0392, Japan

**Keywords:** monolith, polymer, catalyst, microreactor, cross-coupling, flow-through reaction

## Abstract

With the aim of developing efficient flow-through microreactors for high-throughput organic synthesis, in this work, microreactors were fabricated by chemically immobilizing palladium-, nickel-, iron-, and copper-based catalysts onto ligand-modified poly(glycidyl methacrylate-*co*-ethylene dimethacrylate) [poly(GMA-*co*-EDMA)] monoliths, which were prepared inside a silicosteel tubing (10 cm long with an inner diameter of 1.0 mm) and modified with several ligands including 5-amino-1,10-phenanthroline (APHEN), iminodiacetic acid (IDA), and iminodimethyl phosphonic acid (IDP). The performance of the resulting microreactors in Suzuki−Miyaura cross-coupling reactions was evaluated, finding that the poly(GMA-*co*-EDMA) monolith chemically modified with 5-amino-1,10-phenanthroline as a binding site for the palladium catalyst provided an excellent flow-through performance, enabling highly efficient and rapid reactions with high product yields. Moreover, this monolithic microreactor maintained its good activity and efficiency during prolonged use.

## 1. Introduction

High-throughput synthesis is required in the globally competitive life sciences industry, and so far, substantial efforts have been dedicated to the development of high-speed synthesis tools and techniques. In this context, microreactors have received considerable interest in organic synthesis due to their effective and efficient chemical reactions, which benefit from their fast kinetic rates, rapid mass transfer, high yields, short reaction times, cost reduction, and low waste production. Additionally, microreactors facilitate the scale-up of industrial chemical processes and the development of automated solution-phase syntheses. Generally, the most widely used microreactors are based on a microfluidic technology that employs capillary tubes [1,2,3] and microchips/microchannels [4,5,6,7,8,9]; however, in these systems, metal catalysts are difficult to recover and reuse. The presence of residual catalysts in the products can generate problems in the synthesis of functional substrates. To overcome this drawback, approaches such as the loading of metal catalysts onto carbon [10,11] or resins [12,13,14,15,16] have been resorted to. So far, flow-through microreactors prepared by immobilizing metal catalysts onto polymeric supports in a chromatography column are the most efficient microreactor technologies. For example, microreactors prepared using nickel or palladium catalysts supported on microporous silicate [17], styrene-based polymers [18,19,20], and polyurea [21] have been applied to Suzuki−Miyaura cross-coupling reactions. However, the effectiveness of these microreactors still needs improvement in term of mass transfer and stability of the catalytic species loaded onto the surface of the solid support material. Consequently, the development of this type of microreactor remains a challenge.

Fundamentally, a flow-through microreactor consists of two components: a ligand and a polymeric support material [22,23]. The properties of both components are essential for the characteristics and application of the microreactor [24]. The ligand is crucial for the highly selective binding of the metal catalyst, and the polymeric support is essential to improve the mechanical strength, chemical stability, mass transfer, and reusability of the microreactor [25,26,27,28]. The use of organic polymer supports for the immobilization of ligands has been widely investigated. Over the past decade, a single piece of porous material, the so-called “monolith”, has attracted increasing attention and interest due to its high permeability and excellent mass transfer, which allows achieving rapid separations and reactions at high flow rates with minimal loss of column efficiency. By taking advantage of these properties, we achieved rapid separations of proteins and DNA on a time scale of seconds [29,30,31]. Moreover, monoliths have great potential as highly efficient catalyst supports for Suzuki–Miyaura [32,33,34] and Heck cross-coupling reactions [34,35] and other organic synthesis reactions [36,37,38,39,40]. In the present paper, we discuss the challenges encountered in the development of a microreactor using poly(glycidyl methacrylate-*co*-ethylene dimethacrylate) [poly(GMA-*co*-EDMA)] monoliths and their applicability to the Suzuki–Miyaura cross-coupling reaction. We investigated several types of ligands and metal catalysts immobilized in the methacrylate-based monolithic column for an effective cross-coupling reaction.

## 2. Results and Discussion

In a direct effort to develop a flow-through microreactor, we prepared poly(GMA-*co*-EDMA)-based monolithic microreactors containing APHEN, IDA, and IDP as the ligands and Pd(II), Ni(II), Fe(II), and Cu(II) as the metal catalysts, as summarized in Table 1. The poly(GMA-*co*-EDMA) monolith was polymerized in situ within the silicosteel tubing as described in the Materials and Methods section.

### 2.1. Optimization of Chemical Reagents for Flow-Through Cross-Coupling Reactions

To conduct the flow-through cross-coupling reactions, we first focused on the optimization of the reaction solvent using methanol (MeOH), ethanol (EtOH), 1-propanol (1-PrOH), and isopropyl alcohol (IPA) with each concentration of 50% (*v/v*) by employing the flow system depicted in Figure 1. Bromobenzene (**1**) and 4-methylphenylboronic acid (**2**) as substrates and Na_3_PO_4_ as a base were passed through the Pd(II)/APHEN-immobilized monolith microreactor 1 (**M1**) with a residence time of 12 s at room temperature (27 °C). The products were successively collected in a vial and analyzed by HPLC using a homemade reverse phase monolithic column (10 cm long, 1.0 mm i.d.) of poly(LMA-*co*-EDMA) [29] and acetonitrile/water (50:50, *v/v*) as an eluent. As shown in Figure 2a, the reaction solvent of IPA proceeded the highest yield (89%) of the 4-methylbiphenyl (**3**) cross-coupling product, whereas the resulted byproduct of 4,4-dimethylbiphenyl (**4**) is comparable (5–7%) for all solvents at a reaction time of 12 s. Then, using a similar reaction, the IPA concentration was varied from 30 to 70% (*v/v*). As shown in Figure 2b, it was found that the reaction efficiency, indicated by the yield of the cross-coupling product (**3**), greatly improved up to IPA 50% (*v/v*). The yield of the cross-coupling product gradually decreased at IPA > 50% (*v/v*). It seems that when the proportion of water is around 50% (corresponding to IPA concentration of 50%), the solubility of Na_3_PO_4_, as the base, reaches saturation. However, when the proportion of water is lower than 50% (IPA > 50%), the base (Na_3_PO_4_) remains undissolved in the solvent, leading to a decrease in reaction efficiency. In addition, bromobenzene (**1**), as a substrate, is insoluble in water. Therefore, when the proportion of water exceeds 50% (IPA < 50%), its solubility gradually decreases, causing a decrease in reaction efficiency. Then, IPA 50% (*v/v*) was applied for further experiment as the reaction solvent. The use of Na_3_PO_4_ as the base for the Suzuki–Miyaura reaction in this work was adopted from Zhang [41].

Using the flow system (Figure 1) and similar reaction as in Figure 2, we then optimized the mole ratio of cross-coupling reagents and base. The yields of the products, i.e., 4-methylbiphenyl (**3**) as a cross-coupling product and 4,4′-dimethylbiphenyl (**4**) as a byproduct, are given in Table 2. It was found that a cross-coupling product was obtained with a yield close to 100% from the substrate (aryl-X) by adding 4-methylphenylboronic acid (**2**) at a molar ratio of more than 1.2. However, the amount of byproduct **4** increased upon increasing the amount of either 4-methylphenylboronic acid (**2**) or the base. Since the cross-coupling product was obtained almost quantitatively even at stoichiometric molar amounts and considering the formation of the byproduct and the yield of the cross-coupling product, we selected a substrate (aryl–X)/4-methylphenylboronic acid/Na_3_PO_4_ molar ratio of 1:1:1 for the subsequent experiments.

By performing the reaction described in Table 2, we examined the reaction time for the reaction of bromobenzene (**1**) with 4-methylphenylboronic acid (**2**), finding no significant difference in the yields from 12 s to 600 s (Figure 3). Surprisingly, the coupling reaction proceeded smoothly in **M1** and afforded a quantitative yield (89%) of 4-methylbiphenyl (**3**) within 12 s at room temperature. In other microreactors reported so far [42,43,44,45,46], a higher temperature and a longer reaction time were often required to obtain quantitative yields in Suzuki–Miyaura cross-coupling reactions. The rapidity of reaction in this work (within several seconds) could be attributed to the high surface area of the poly(GMA-*co*-EDMA) monolith as a polymeric support of the metal catalyst, which can provide improved kinetic rates. In addition, room-temperature transformation is a highly desirable process from a synthetic point of view. This phenomenon can be explained by the pore size distribution of the poly(GMA-co-EDMA) monolith used in this work. Since we used a similar composition, ratio, and size of the silicosteel tubing to prepare the poly(GMA-co-EDMA) monolith as in our previous work [24], we assume that the present monolith has similar characteristics to those of the product obtained therein, such as total (ε_t_), external (ε_e_), and internal (ε_i_) porosities of 0.63, 0.25, and 0.39, respectively. This monolith exhibits predominantly a mesopore character because the internal porosity is higher than the external porosity, which represents the flow-through pore character. We determined the pore size distribution of the monolith to be 33.8% macropores/flow-through pores (>50 nm, 63.3% mesopores (2–50 nm), and 2.9% micropores (<2 nm). Mesopores in a range of 5–15 nm were favorable (51.2%), and the volume fraction for flow-through pores (>300 nm) was 23.7%. In general, to achieve a favorable mass transfer in monoliths, the volume of mesopores should be minimized and the macropores/flow-through pores should be maximized. It should be noted, however, that surface interactions are mainly dependent on the mesopores and micropores. A large surface area (represented by mesopores) is required to achieve an appropriate exposure of the ligands for postchemical modification and an adequate binding of the catalysts [47]. Mesopores are preferred over micropores because micropores might result in adverse irreversible adsorption and high flow resistance. As a result, the monolith support must have sufficient porosity to reach a balance between flow-through pores (macropores) and a large surface area (mesopores) to achieve excellent ligand and catalyst binding for the intended application. In addition, the reaction space, represented by the size of flow-through pores in our previous poly(GMA-*co*-EDMA) monolith, is several to several tens of micrometers [31]. Therefore, diffusion distance is quite short, and then the substrates are effectively collided and react with each other in a small space existing immobilized Pd(II). Furthermore, an irregular flow channel will accelerate the agitation [48,49]. Thus, the catalytic reaction would be effectively carried out in this Pd(II)-immobilized monolithic microreactor (**M1**).

### 2.2. Effect of the Metal Catalyst and Ligands on the Efficiency of Flow-Through Cross-Coupling Reactions

Next, we investigated the effect of the metal catalyst by evaluating Pd(II), Ni(II), Fe(II), and Cu(II) catalysts binding to the **APHEN** ligand in the **M1**–**M4** microreactors according to the reaction depicted in Table 2. As shown in Figure 4, **M1** provided superior performances compared with **M2**–**M4**. Assuming that the reaction proceeds via a similar mechanism for all metal catalyst/**APHEN**-immobilized poly(GMA-*co*-EDMA) monoliths, it can be concluded that the Pd(II) catalyst exhibited the highest reaction rate because it afforded the highest yield of 4-methylbiphenyl (**3**) within a short time. Generally, Pd(0) is regarded as the catalytically active species in Pd catalysis. In contrast, Ni(II), Fe(II), and Cu(II) are not so readily reduced by the base, solvent, or ligand to the corresponding metal(0) species, which could be invoked to explain the lower reaction rates and yields obtained with the Ni(II)-, Fe(II)-, and Cu(II)-immobilized **M2**–**M4** microreactors. According to previous reports showing that the use of Zn(II) as a reductant in the Ni(II)-catalyzed Suzuki–Miyaura reaction promotes the conversion from Ni(II) to Ni(0) at elevated temperature (80 °C–130 °C) [50,51], the addition of reductants could be expected to improve the result of the **M2**–**M4** system. However, such a high temperature is not conducive to the present coupling reaction conducted at room temperature.

We also examined the efficiency of the poly(GMA-*co*-EDMA)-based microreactors possessing different ligands (**M1**, **M5**, **M6**) and Pd(II) as a catalyst in the Suzuki–Miyaura reaction of bromobenzene (**1**) with 4-methylphenylboronic acid (**2**). As can be seen in Figure 5, the yield of **3** in **M6** (IDP ligand) increased with increasing the reaction time, reaching 80% at 600 s. In contrast, **M1** (APHEN ligand) and **M5** (IDA ligand) afforded **3** in a higher yield (89%) with a faster reaction rate. **M1** outperformed **M5** slightly because it produced **3** in a higher yield at a reaction time of ≤60 s, although both microreactors generated comparable yields at longer reaction times. Additionally, the production of byproduct **4** in **M5** increased slightly with increasing the reaction time. We attribute the different results obtained with **M1**, **M5**, and **M6** to the different binding of Pd(II) to each ligand. Thus, Pd(II)-IDP forms an eight-membered ring complex that is less stable than the five- and six-membered rings of Pd(II)-APHEN and Pd(II)-IDA, respectively. Therefore, Pd(II) is adsorbed more strongly in **M1** and **M5** than in **M6**. In addition, the –NH_2_ group of APHEN is more reactive than the –NH group of IDA, which could result in a higher amount of APHEN being attached to the monolith support via the ring-opening of the electrophilic epoxy groups. To prove our hypothesis, we analyzed the amounts of Pd in the microreactors by ICP-MS. For this experiment, the monoliths inside the **M1**, **M5**, and **M6** microreactors were removed from the column by applying a high pressure of over 20 MPa, dried at 105 °C for 24 h, and subsequently digested via microwave-assisted acid digestion. We found that the Pd content in **M1**, **M5**, and **M6** was 250, 180, and 80 nmol, respectively. These amounts correspond to the amount of ligand in the microreactors because a 1:1 metal–ligand complex is formed in each case, which confirms our hypothesis.

Having selected **M1** as the best microreactor, we then explored the substrate scope of the present Suzuki–Miyaura cross-coupling reaction. We initially chose 4-methylphenylboronic acid (**2**) as the common electrophilic coupling partner of aryl halides. As shown in Table 3, 4-methylphenylboronic acid (**2**) reacted with aryl halides to afford the corresponding biphenyl **3** in good to excellent conversions (Table 3, entries 1–3) within 12 s at room temperature. In contrast, aryl chloride afforded a poor result due to its weak reactivity compared with that of aryl iodide, bromide, and trifluoromethyl sulfonate (Table 3, entry 4).

Then, we evaluated other types of electrophilic substrates having electron-withdrawing and donating groups. Aryl bromides possessing an –NO_2_ or –COOH group at *para* position produced excellent yields (>99%) of the corresponding biphenyls **5** and **6** within 60 s (Table 3, entries 5 and 6), and a fairly good conversion was achieved with 4-hydroxy bromobenzene (Table 3, entry 7) after 600 s. As the electron-withdrawing groups, the nitro and carboxyl stabilized carbocation in the reactions, while the hydroxyl group donated electrons, making the carbocation less stable.

To further examine the suitability of **M1**, we conducted the reactions of sterically hindered substrates of bromobenzene derivatives with 4-methylphenylboronic acid (**2**). Substrates bearing electron-donating groups at *meta* (Table 3, entry 8) and *para* (Table 3, entry 9) positions smoothly underwent the coupling reaction to afford complete conversion of the desired biphenyl products **8** and **9** within only 60 s. The presence of a longer chain in the donating-electron group at *para* position (Table 3, entry 10) also resulted in an excellent yield of biphenyl **10** within 600 s. However, for *ortho-*substituted substrates, the conversions decreased to 75% (Table 3, entry 11) and 52% (Table 3, entry 12) at a reaction time of 600 s due to steric hindrance.

We also examined the chemical stability of Pd(II)-immobilized monolithic microreactor via APHEN ligand (**M1**) by passing bromobenzene (**1**) through the microreactor at a reaction time of 12 s for at least 20 cycles. As shown in Figure 6, the yield of a cross-coupling product, 4-methylbiphenyl (**3**), was 89.08 ± 0.76% (A1) with and the retention time of 5.46 ± 0.66 min (A2) whereas the yield and retention time of a byproduct, 4,4′-dimethylbiphenyl (**4**), were 7.16 ± 0.34% (B1) and 7.21 ± 0.10 min (B2), respectively. All reaction products and retention times after 20 cycles do not change significantly since the relative standard deviations (RSDs) are less than 5%. Additionally, we also thought that a small amount of Pd had been eluted, and we tried to determine Pd in the effluent by ICP-MS, but it was below the detection limit (not detected). These results indicated that the monolithic microreactor **M1** provides good chemical stability and efficiency for cross-coupling reactions.

The thermal stability of the microreactor **M1** was examined by introducing bromobenzene (**1**) with 4-methylphenylboronic acid (**2**) in the reaction temperature range of 27 ^o^C to 80 °C. As shown in Figure 7, the results showed no significant change of the 4-methylbiphenyl cross-coupling product (**3**) and the 4,4′-dimethylbiphenyl byproduct (**4**) at all reaction temperatures examined, indicating good thermal stability of the microreactor. Additionally, from a synthetic point of view, it is advantageous because this microreactor can work at room temperature and gives good results for cross-coupling reactions.

## 3. Materials and Methods

### 3.1. Reagents

All chemicals were used as received without further purification. For the preparation of poly(GMA-*co*-EDMA) monoliths inside a silicosteel tubing, glycidyl methacrylate (GMA), ethylene dimethacrylate (EDMA), 1-propanol, 1,4-butanediol, and 2,2′-azobis(isobutyronitrile) (AIBN) were purchased from Wako Pure Chemicals (Osaka, Japan), and 3-methacryloxypropyltrimethoxysilane (MPS) was supplied by Sigma-Aldrich (Tokyo, Japan). The ligands 5-amino-1,10-phenanthroline (APHEN), iminodiacetic acid (IDA), and iminodi(methyl phosphonic acid) (IDP) were obtained from Sigma-Aldrich Japan (Tokyo, Japan). PdCl_2_.(CH_3_CN)_2_ of the highest pure grade commercially available was purchased from Sigma-Aldrich Japan. Similarly, NiCl_2_, FeCl_2_, and CuCl_2_ were also obtained from the same company. For the Suzuki–Miyaura cross-coupling reactions, iodobenzene, bromobenzene, chlorobenzene, phenyl trifluoromethanesulfonate, 4-bromonitrobenzene, 4-bromobenzoic acid, 4-bromophenol, 2-bromotoluene, 3-bromotoluene, 4-bromotoluene, 4-pentylbromobenzene, 2,6-dimethylbromobenzene, 4-methylbiphenyl, and 4,4′-dimethylbiphenyl were purchased from Tokyo Kasei (Tokyo, Japan). Na_3_PO_4_ was obtained from Sigma-Aldrich Japan, and isopropanol and 4-methylphenylboronic acid were purchased from Wako Pure Chemicals. As organic solvents, acetonitrile, tetrahydrofuran, methanol, dimethyl sulfoxide (DMSO), and acetone were also supplied by Wako Pure Chemicals. Water used in all the experiments was purified by an ELIX 10/Milli-Q Element A-10 purification system (Millipore).

### 3.2. Procedure for the Preparation of Flow-Through Monolithic Microreactors

The preparation of the microreactors comprised the following three steps: (a) preparation of the poly(GMA-co-EDMA) monolith as a polymer base, (b) chemical reaction of the ligands with the epoxy group of the organic polymer monolith, and (c) immobilization of the catalyst by binding it to the ligand. In situ copolymerization inside a silicosteel tubing (1.02 mm i.d., 1/16 o.d., GL Sciences, Tokyo, Japan) was used to create the poly(GMA-co-EDMA) monolith. According to a methodology outlined in our previous work [29], to provide olefins on the inner tube wall of the silicosteel tubing for covalently binding the organic polymer monolith, the tubing was first silanized with MPS. After that, the treated tubing was cut into the necessary length (100 mm long in this study). All of the polymerization mixture’s components were transferred to a glass vial, composing of 6000 μL GMA, 2000 μL EDMA, 7000 μL 1-propanol, 4000 μL 1,4-butanediol, 1000 μL water, and AIBN as an initiator (1 wt. percent of the total monomer). Before filling the silanized silicosteel tube, the resulting solution (the mixture of monomers, porogen, and initiator) was ultrasonically homogenized for 10 min. The tube was then placed in an oven for 24 h at 60 °C after both ends were sealed with stainless fittings to complete the polymerization process. The poly(GMA-co-EDMA) monolith was washed with ethanol and water using a high-pressure liquid chromatography (HPLC) pump to remove the leftover porogenic solvent and unreacted monomers inside the monolithic column. Subsequently, the ligands (APHEN, IDA, and IDP) were chemically functionalized to the polymeric support by ring-opening the monolith’s epoxy groups by passing each 0.1 mmol ligand solution in DMSO (1 mL) through the poly(GMA-co-EDMA) monolithic column using a syringe pump (MSP-DT2 model, As One ltd, Osaka, Japan) with a flow rate of 5 μL/min. The column was then placed in the oven at 60 °C for 14 h after both ends were sealed with stainless fittings. Before further treatment, the column was washed for 1 h with ethanol and water using an HPLC pump with a flow rate of 50 μL/min. Then, 0.1 mmol solutions of Pd(II), Ni(II), Cu(II), and Fe(II) in acetone (2 mL) were retained on the ligand-immobilized monolithic support by passing the catalyst solution through the monolithic column using the syringe pump with a flow rate of 5 μL/min at room temperature (27 °C). In the case of NiCl_2_, this compound was dissolved in acetone/water with a ratio of 4:1 (*v/v*). All monolithic microreactors were washed for 1 h with acetone and water before use at flow rate of 50 μL/min. At this flow rate, the pressure drop of all microreactors was in the range of 0.6–0.7 MPa using water as a solvent.

### 3.3. Instrumentation and Measurement

The flow system for the evaluation of the microreactors consisted of a Shimadzu modular system equipped with two HPLC pumps (LC-20AD), a communication bus module (CBM-20A), and a column oven (CTO-20AC), as shown in Figure 1a. Flow lines were made of PEEK tubing (0.13 mm i.d., 1/16” o.d.), and T-pieces made of PEEK were used as a mixing joint. The Suzuki–Miyaura cross-coupling reagents (R1, R2) with a concentration of 10 mM were introduced into the flow system and passed through the monolithic microreactor at various flow rates ranging from 8 μL/min to 390 μL/min, which correspond to reaction times of 12–600 s inside the monolithic microreactor, to form the biphenyl products. After the reaction product was collected in a small vial, it was immediately analyzed by injecting 1 μL of the effluent into the HPLC system (Figure 1b), which was equipped with pumps and a communication bus module similar to that of Figure 1a, a Rheodyne 8125 injector with a homemade 1 μL sample loop, a model SPD-20A UV/VIS detector with a semimicro flow cell (2.5 μL), and a homemade reverse phase column of poly(lauryl methacrylate-*co*-ethylene dimethacrylate) [poly(LMA-*co*-EDMA)] [29]. The separation of the products was conducted under isocratic elution by employing acetonitrile/water (50:50 *v/v*) as a mobile phase at room temperature and a flow rate of 0.05 mL/min. System control and data acquisition were performed using the Shimadzu LC solution software.

To determine the catalyst amount that was immobilized into the microreactor, the monoliths were removed from the column by applying a high pressure of over 20 MPa followed by drying at 105 °C for 24 h. These samples were then weighed, transferred into a 100 mL poly(tetrafluoroethylene) vessel, and subjected to acid digestion using a mixture of 60% ultrapure HNO_3_ (4 mL) and 30% ultrapure H_2_O_2_ (1 mL) in an ETHOS E Microwave Extraction System (Milestone General, Italy). The microwave heating program for the digestion was as follows: The temperature increased from room temperature to 70 °C within 2 min at 1000 W. For safety considerations, the temperature was reduced to 50 °C (0 W) in 3 min before heating continually to 200 °C (1000 W) over a 10 min period. The temperature and power were then maintained at this level for 25 min, followed by a 60 min cooling period inside the oven. This procedure was repeated twice. Then, the concentration of the catalyst in the digested samples, which contained an internal standard solution of ^89^Y (10 ppb), was determined via high-resolution inductively coupled plasma mass spectrometry (ICP-MS; ELEMENT2, Thermo Fisher Scientific, USA) at the middle-resolution mode (R = 4000).

## 4. Conclusions

In conclusion, we fabricated flow-through microreactors using a poly(GMA-*co*-EDMA) monolith prepared by in situ copolymerization of GMA and EDMA as the polymer support. Subsequently, several kinds of ligands and metal catalysts were chemically immobilized onto the monolith. The **M1** microreactor, which consisted of the poly(GMA-*co*-EDMA) monolith chemically attached with **APHEN** as a binding site to the Pd(II) catalyst, provided an excellent flow-through performance in Suzuki−Miyaura cross-coupling reactions, enabling highly efficient (high yield) and rapid syntheses (from 12 to 600 s at room temperature). Moreover, this microreactor maintained its good activity and efficiency after prolonged use. The present monolithic microreactor has great potential for flow-through and high-speed cross-coupling organic synthesis.

## Figures and Tables

**Figure 1 molecules-26-07346-f001:**
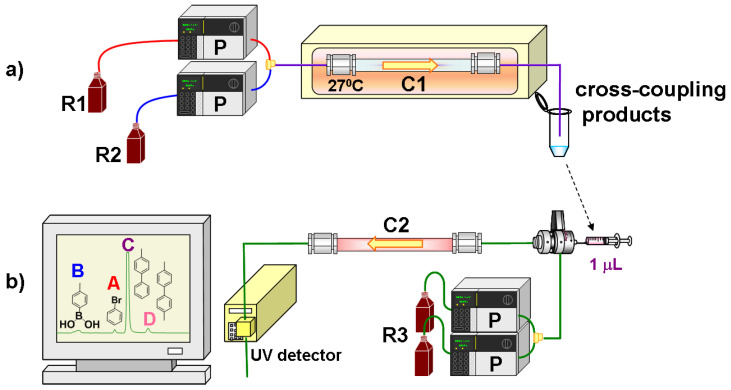
(**a**) Flow-through microreactor system for Suzuki–Miyaura cross-coupling reactions and (**b**) HPLC system for the analysis of cross-coupling products. R1: aryl–X, R2: 4-methyl boronic acid and Na_3_PO_4_, R3: acetonitrile/water (50:50 *v/v*), P: HPLC pump, C1: monolithic microreactors **M1**–**M6** (1 mm i.d. × 100 mm length), C2: poly(lauryl methacrylate-*co*-ethylene dimethacrylate) monolithic column (1 mm i.d. × 100 mm length).

**Figure 2 molecules-26-07346-f002:**
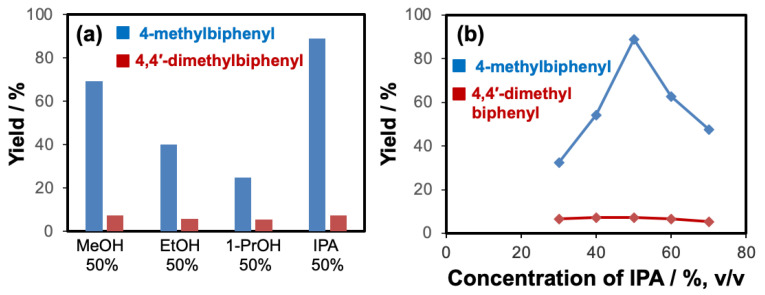
Effect of solvent (**a**) and (**b**) concentration of IPA on the reaction of bromobenzene with 4-methylphenylboronic acid using the microreactor **M1**. Reaction time inside **M1** was 12 s, and reaction temperature was 27 °C. MeOH: methanol, EtOH: ethanol, 1-PrOH: 1-propanol, IPA: isopropyl alcohol. Mole ratio of bromobenzene/4-methylphenylboronic acid/Na_3_PO_4_ = 1:1:1.

**Figure 3 molecules-26-07346-f003:**
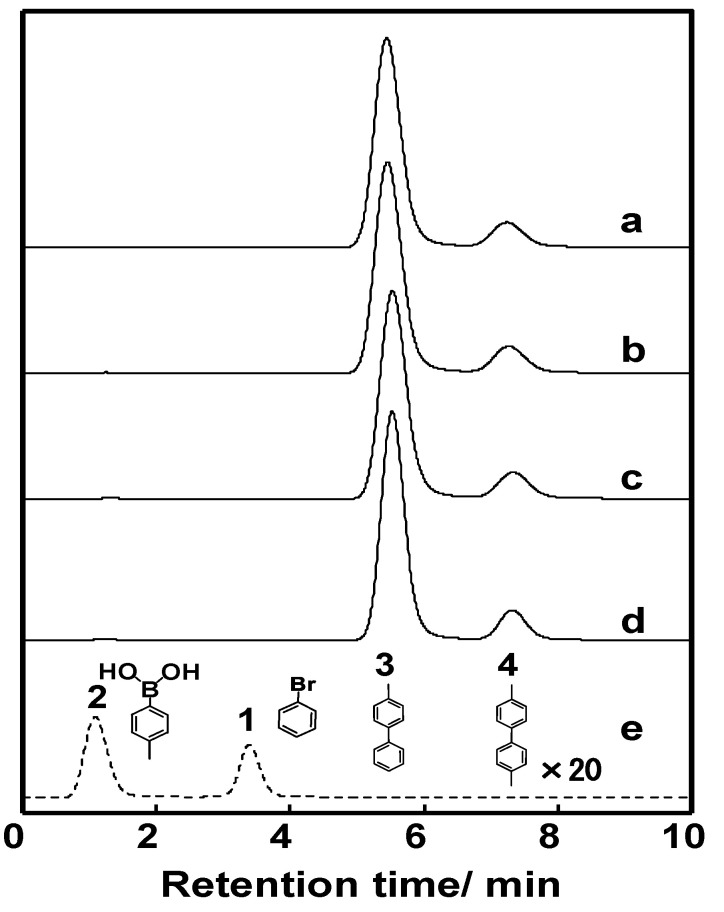
Chromatogram of the cross-coupling products of the Suzuki–Miyaura reaction at different reaction times. Mole ratio of bromobenzene/4-methylphenylboronic acid/Na_3_PO_4_ = 1:1:1, microreactor: **M1,** reaction times: 12 s (**a**), 60 s (**b**), 300 s (**c**), 600 s (**d**), 0 s (**e**), reaction temperature: 27 °C.

**Figure 4 molecules-26-07346-f004:**
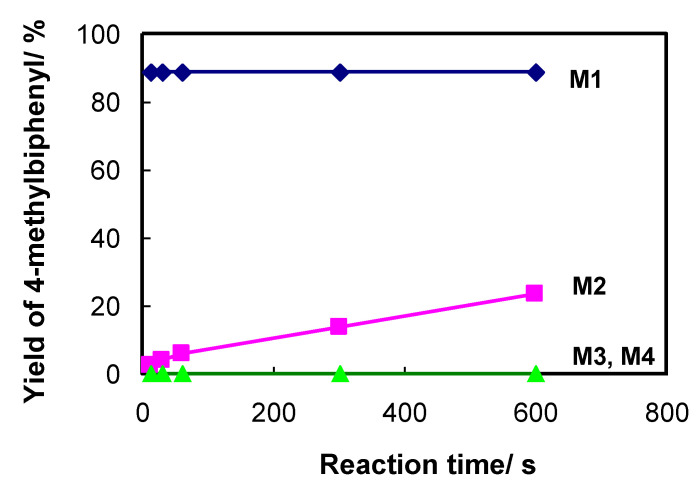
Effect of metal catalysts on the reaction of bromobenzene with 4-methylphenylboronic acid. Other conditions were similar to those in Figure 3.

**Figure 5 molecules-26-07346-f005:**
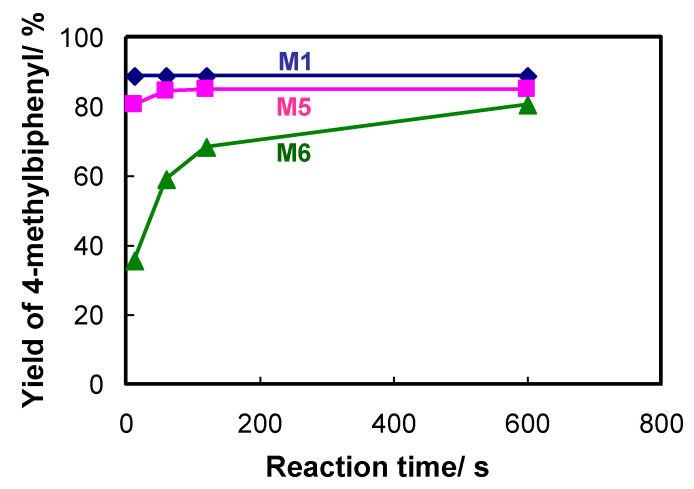
Effect of ligands on the reaction of bromobenzene with 4-methylphenylboronic acid. Other conditions were similar to those in Figure 3.

**Figure 6 molecules-26-07346-f006:**
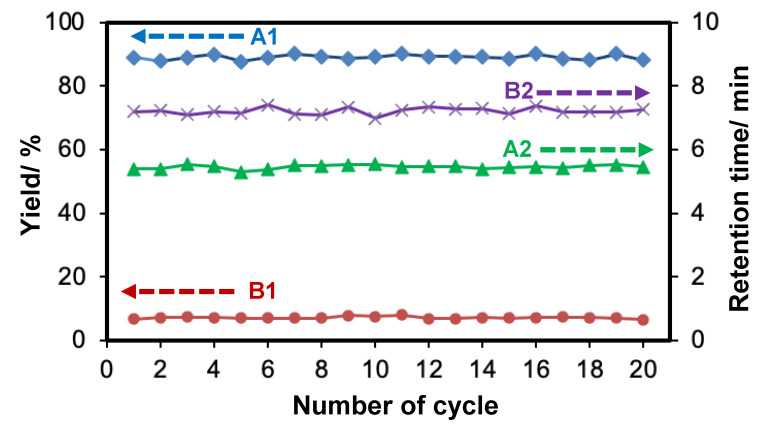
The chemical stability test of Pd(II)-immobilized monolithic microreactor (**M1**) through the reaction of bromobenzene with 4-methylphenylboronic acid. Reaction time inside **M1** was 12 s. Other conditions are simar as in Figure 3. A1 and A2 are the cross-coupling product and retention time of 4-methylbiphenyl, respectively. B1 and B2 are the byproduct and retention time of 4,4′-dimethylbiphenyl.

**Figure 7 molecules-26-07346-f007:**
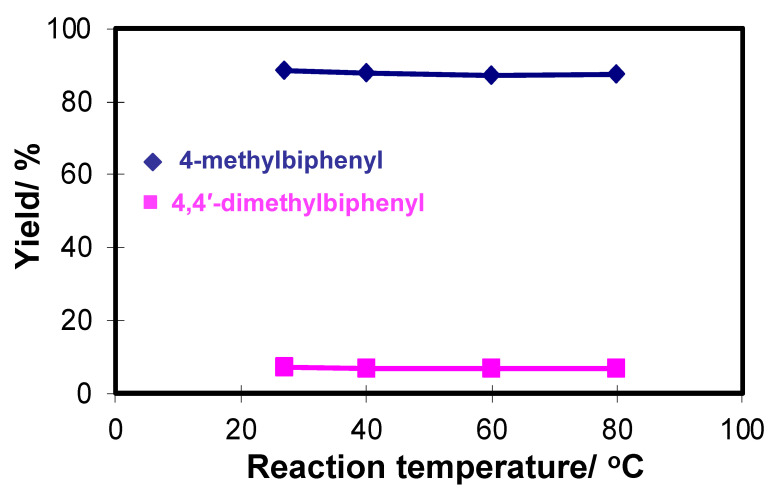
The thermal stability test of the microreactor M1 at various temperatures for the reaction of bromobenzene with 4-methylphenylboronic acid. Reaction time inside M1 was 12 s. Mole ratio of bromobenzene/4-methylphenylboronic acid/Na_3_PO_4_ = 1:1:1.

**Table 1 molecules-26-07346-t001:** Metal-immobilized Poly(GMA-*co*-EDMA)-based monolithic flow-through microreactors.

Polymeric Support	Ligand	Catalyst	Microreactor
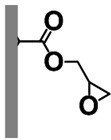 **Poly(GMA-*co*-EDMA)** **Monolith**	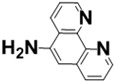 **APHEN**	Pd(II)	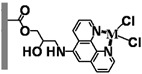 **M1** = Pd(II), **M2** = Ni(II) **M3** = Fe(II), **M4** = Cu(II)
		Ni(II)	
		Fe(II)	
		Cu(II)	
	HN(CH_2_COOH)_2_ **IDA**	Pd(II)	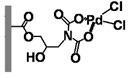 **M5**
	HN(CH_2_PO_3_H_2_)_2_ **IDP**	Pd(II)	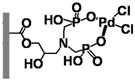 **M6**

**Table 2 molecules-26-07346-t002:**
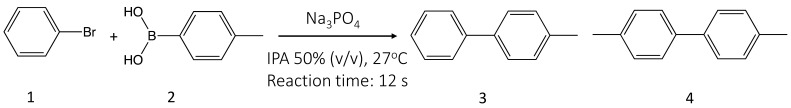
Optimization of the mole ratio of cross-coupling reagents and base.

Mol 1	Mol 2	Mol Na_3_PO_4_	Yield 3 (%)	Yield 4 (%)
1	1	0.1	7	1
1	1	0.5	50	5
1	1	1	89	7
1	1	1.5	90	10
1	1.2	1	97	9
1	1.5	1	99	11

IPA: isopropyl alcohol; microreactor: Pd(II)-immobilized monolith poly(GMA-*co*-EDMA) (**M1**).

**Table 3 molecules-26-07346-t003:**
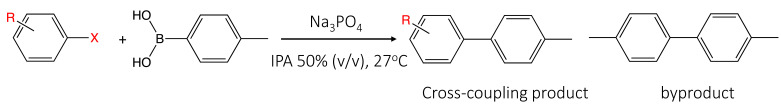
Substrate (Aryl–X) scope of the coupling reaction.

Entry	Aryl–X	Cross-Coupling Product	Yield (%), [] Indicates the Yield of Byproduct
12 s *^a^	60 s *^a^	300 s *^a^	600 s *^a^
1	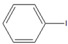	** 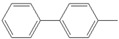 ** **3**	93	[4]	-		-		-	
2	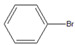	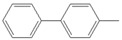 **3**	89	[7]	-		-		-	
3	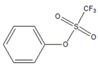	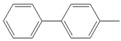 **3**	74	[13]	-		-		-	
4	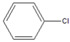	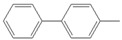 **3**	2	[16]	-		-		-	
5	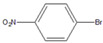	** 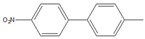 ** **5**	93	[4]	99		-		-	
6	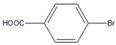	** 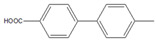 ** **6**	88	[6]	99		-		-	
7	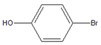	** 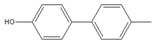 ** **7**	46	[8]	50	[9]	62	[14]	69	[15]
8	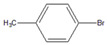	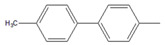 **8**	95		99		-		-	
9	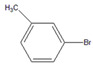	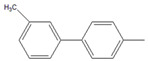 **9**	84	[6]	99		-		-	
10	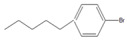	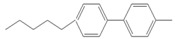 **10**	60	[7]	63	[10]	74	[13]	99	
11	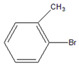	** 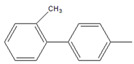 ** **11**	47	[4]	55	[5]	69	[7]	75	[9]
12	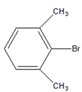	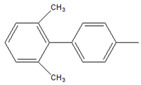 **12**	12	[3]	21	[4]	39	[8]	52	[12]

*^a^ Reaction time; IPA: isopropyl alcohol; microreactor: Pd(II)-immobilized poly(GMA-*co*-EDMA) monolith (**M1**).

## Data Availability

The data in this study are available upon request from the corresponding authors.

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
