# Peer review of "Preparation of Metal-Immobilized Methacrylate-Based Monolithic Columns for Flow-Through Cross-Coupling Reactions"

_molecules, 2021, doi:10.3390/molecules26237346_

Round 1
Reviewer 1 Report
This manuscript provides a method to conduct heterogeneous catalysis in microreactors for the synthesis of fine chemicals. The structure and contents of this manuscript seem reasonable. However, the authors need to add more discussion for the experimental phenomena before its publication.
- Lots of relevant references are missing in this manuscript. For example, in this sentence “In other microreactors reported so far, a higher temperature and a longer reaction time were often required to obtain quantitative yields in Suzuki–Miyaura cross-coupling reactions”, no any reference is not cited.
- Why could the authors get much higher reaction rate for the Suzuki–Miyaura cross-coupling reactions even compared with homogeneous reaction processes? They authors should explain this phenomena clearly beside the reason based on the high surface area of the poly(GMA-co- 126 EDMA) monolith.
- For the chemical stability of the metal-immobilized methacrylate-based monolithic columns, the authors should show a figure to describe it.
- How about the pressure drop in such kinds of microreactors?
Reviewer 2 Report
This manuscript shows an interesting and important application of microreactors. The work involves a series of metal-immobilized ligand-modified poly(GMA-co-EDMA) monoliths and provides a detailed evaluation of Suzuki-Miyaura reaction using systematic experiments. The manuscript also shows the effect of reaction time, mole ratio of reagents and base, metal type, ligand type and substrate type on the product yield. I recommend accepting this manuscript for molecules only after a minor revision.
- It was mentioned that the reaction happened at room temperature, so when the products were collected in the vial, the reaction would continue before analysis, this begs the question that whether the measured product yields reflected the real reaction performance in the microreactor, especially for the case with short reaction time. I think the end effect should be taken into consideration in a proper way.
- The M1 microreactor provided an excellent performance with extremely low concentrations of the reagents, I wonder whether it could maintain a good performance when increasing the concentrations of the reagents.
- How is the thermal stability the microreactor?
Reviewer 3 Report
This manuscript described an efficient heterogeneous for Suzuki -Miyaura coupling reactions in the microreactors based on the poly(GMA-co-EDMA) monolith, where several types of ligands with different immobilized metals were grafted. The effect of coordinating ligands and metals were evaluated and explained. The catalyst showed quantitative yield of product with ultra fast time at room temperature, which paves the way for high-speed cross-coupling organic synthesis. I suggested the manuscript to be accepted with minor changes. Here're the suggestions below:
- In line 80, optimization of the reaction condition including choosing base and solvent should be included.
- Line 129, the high reactivity was ascribed to the proper distribution of mesopores. Please provide some literature support. How is it compared with homogeneous PHEN-Pd complex? Because the bipyridine based ligands are already highly reactive.
- Table 2, it would be better to point that M1 is packed in the column as catalyst. Also what does 50% IPA mean and what is the other eluent?
- Line 274, when screening different substrates, why fair withdrawing hydroxyl substituent has low yield while strong withdrawing group like nitro has higher yield?
- Line 296, the catalyst show high recyclability with at least 20 cycles. What is the turnover number based on the mole of Pd?
- After the reaction, it is better to analyze the Pd loading to confirm the maintaining of the coordination and no metal leaching.
